# Method for the Production and Purification of Plant Immuno-Active Xylanase from *Trichoderma*

**DOI:** 10.3390/ijms22084214

**Published:** 2021-04-19

**Authors:** Gautam Anand, Meirav Leibman-Markus, Dorin Elkabetz, Maya Bar

**Affiliations:** 1Department of Plant Pathology and Weed Research, Plant Protection Institute, Agricultural Research Organization, Volcani Institute, Rishon LeZion 50250, Israel; gautaming@gmail.com (G.A.); meiravleibman@gmail.com (M.L.-M.); dorin.elkabetz@mail.huji.ac.il (D.E.); 2Department of Plant Pathology and Microbiology, Faculty of Agriculture, Hebrew University of Jerusalem, Rehovot 91905, Israel

**Keywords:** xylanase, innate immunity, enzyme purification

## Abstract

Plants lack a circulating adaptive immune system to protect themselves against pathogens. Therefore, they have evolved an innate immune system based upon complicated and efficient defense mechanisms, either constitutive or inducible. Plant defense responses are triggered by elicitors such as microbe-associated molecular patterns (MAMPs). These components are recognized by pattern recognition receptors (PRRs) which include plant cell surface receptors. Upon recognition, PRRs trigger pattern-triggered immunity (PTI). Ethylene Inducing Xylanase (EIX) is a fungal MAMP protein from the plant-growth-promoting fungi (PGPF)–*Trichoderma*. It elicits plant defense responses in tobacco (*Nicotiana tabacum*) and tomato (*Solanum lycopersicum*), making it an excellent tool in the studies of plant immunity. Xylanases such as EIX are hydrolytic enzymes that act on xylan in hemicellulose. There are two types of xylanases: the endo-1, 4-β-xylanases that hydrolyze within the xylan structure, and the β-d-xylosidases that hydrolyze the ends of the xylan chain. Xylanases are mainly synthesized by fungi and bacteria. Filamentous fungi produce xylanases in high amounts and secrete them in liquid cultures, making them an ideal system for xylanase purification. Here, we describe a method for cost- and yield-effective xylanase production from *Trichoderma* using wheat bran as a growth substrate. Xylanase produced by this method possessed xylanase activity and immunogenic activity, effectively inducing a hypersensitive response, ethylene biosynthesis, and ROS burst.

## 1. Introduction

Plants are sessile and cannot escape environmental stressors or pathogens. In order to cope with environmental stress conditions (both biotic and abiotic), plants have developed sophisticated mechanisms for sensing different stresses and adapting to them by rapid, dynamic, and complex physiological changes.

Plants have developed an immune system that is not acquired but innate for protection against pathogens. The innate immune system in plants is based on complex and effective defense mechanisms [1]. The induced responses include rapid and immediate responses such as alkalinization and a burst of active oxygen and nitrogen compounds, which occur within seconds to minutes from the moment the microorganism is detected [2,3]. Moreover, there are complex and prolonged responses, including alteration of the gene expression pattern in the plant [4,5]. Plant defense responses are triggered by elicitors such as microorganism associated molecular patterns (MAMPs). MAMP molecules have been isolated from a wide variety of pathogenic and non-pathogenic microorganisms [6]. These components are recognized by sensory pattern recognition receptors (PRRs). Following detection, the PRR components activate pattern-triggered immunity (PTI) [7]. To probe plant defense and attempt to understand plant signaling mechanisms, one of the MAMPs employed in scientific studies is xylanase [8,9,10,11].

Xylanases are enzymes of the hemicellulolytic complex that act on the central chain of hemicellulose. Hemicellulose consists of xylan (a heteropolysaccharide substituted with monosaccharides such as l-arabinose and glucuronic acid ), d-mannoses, and organic acids. There are two types of xylanases: the endo-1, 4-β-xylanases (EC 3.2.1.8), that act inside the xylan structure and release xylooligosaccharides (XOS) as the main product, and the β-d-xylosidases (EC 3.2.1.37) that externally hydrolyze the xylan chain and small XOS, thus releasing xylose [12,13]. These enzymes are produced mainly by microorganisms and take part in the breakdown of plant cell walls during the process of plant colonization. Thus, it is not surprising that the plant is able to recognize them as MAMPs and activate their defenses in response. Immune pathways activated by xylanase have been elucidated in several cases [9,10,11,14].

One notable plant-immunity-inducing xylanase is the family 11 endo-1,4-β-xylanase present in fungi of *Trichoderma* spp., also termed ethylene inducing xylanase (EIX) [15]. *Trichoderma* is a plant-growth-promoting fungi (PGPF) and biocontrol agent known to promote growth and prime plant immunity [16]. The EIX protein can act as an MAMP and elicit plant defense responses in several cultivars of *Solanaceae*, including tobacco (*Nicotiana tabacum*) and tomato (*Solanum lycopersicum*) [8,17]. EIX elicits a hypersensitive response (HR) and various defense responses, including reactive oxygen species (ROS) burst, pathogen-related (PR) gene expression, extensive electrolyte leakage, and, as indicated by its name, induces ethylene biosynthesis [8,10]. The response to EIX is controlled by a single dominant locus termed *LeEIX* [17]. This locus contains two genes (*LeEIX1* and *LeEIX2*) encoding leucine-rich repeat receptor-like proteins (LRR-RLPs). Both LeEIX1 and LeEIX2 gene products are capable of binding the EIX elicitor independently. However, only LeEIX2 is able to transmit the signal necessary to induce defense responses [18]. The combination of a protein from a non-pathogenic microorganism that is able to trigger plant defense responses without causing disease, together with the fact that its plant immune-receptors are known, makes it an excellent tool in the studies of plant immunity.

Xylanases have gained importance due to their industrial applications in the food and feed industries. For example, xylanase from *Penicillium occitanis* is used in the bread-making process [19]. Xylanases are used for the clarification of fruit juices [20]. Xylanases have also gained importance as an animal feed supplement, contributing to the digestibility of fibrous foods and allowing for effective utilization of difficult-to-process nutrients [21]. Xylanases are also used in paper and pulp industries for bio-bleaching and de-inking processes [22,23].

Among the eukaryotic microbial sources for xylanase purification, filamentous fungi are especially interesting as they secrete these enzymes in liquid cultures. Their xylanase levels are exceedingly high when compared to those found in yeasts [24].

Currently, there is a lack of cost-effective commercially available sources of xylanases that retain plant immunogenic activity. Due to important applications for microbial xylanases in scientific research, there is a need to identify an efficient and safe source and production strategy for these enzymes. This problem led us to the present study, with the aim of producing the xylanase EIX in an efficient, low-cost, and easily accessible method. We used the fungus *Trichoderma harzianum* for xylanase production under submerged fermentation (SmF) conditions. *Trichoderma* has a generally safe (GRAS) status, and it is well known for its biocontrol activity. Moreover, it can produce and extracellularly secrete enzymes in different media [25]. Wheat bran, which is an agro-waste, was used as a substrate for xylanase production. In this article, we present a cost and yield-effective production strategy for fungal xylanases using agricultural waste as a substrate [26].

## 2. Results

Several works suggest that SmF is the preferred strategy for xylanase production from bacteria and fungi. Approximately 90% of global commercial xylanase is produced through SmF [27,28]. In this work, xylanase was produced under SmF using wheat bran as a substrate. Xylanase production was confirmed by enzyme activity assay. The crude enzyme was harvested on the fifth day of fermentation and filtered. The clear culture filtrate was centrifuged and subjected to ammonium sulfate precipitation (0–30%, 30–60%, 30–40%, 40–50%, and 50–60% saturation).

We evaluated the degree of purification and verified the presence of EIX protein in the different ammonium sulfate fractions. We compared crude extract, fractions obtained after 30–60% ammonium sulfate saturation, and after 40–50% ammonium sulfate saturation, to a positive control (PC) (Fluka, Milwaukee, WI, USA—now discontinued). Equal amounts were loaded onto SDS-PAGE. EIX is ~22 kDa [8], and indeed, in all lanes, we are able to detect a band of approximately 22 kDa that is suspected to be EIX (Figure 1a). In addition to this band, we are able to detect, in the lane of the positive control, two other bands of ~34 and 14 kDa, which could be a result of aggregation and breakage of the EIX protein, respectively. Alternatively, they could be a result of the contamination of unrelated proteins. The ~34 kDa band was present in all the samples, while the ~14 kDa was observed in the positive control, the 30–60% fraction, and very faintly in the crude fraction (Figure 1a). Additionally, in all of the fractions, we observed several higher bands (Figure 1a). In order to verify that the ~22 kDa band is indeed EIX, we blotted the gel onto nitrocellulose membranes and incubated it with EIX-specific polyclonal antibodies [29]. As shown in Figure 1b, the ~22 kDa band was recognized by the anti-EIX antibodies, proving it to be EIX. As can be seen in the positive control, and very faintly in the 30–60% fraction, the lower ~14 kDa band was also recognized by anti-EIX antibodies (Figure 1b). We thus concluded that the 14 kDa is a peptide fragment of EIX, and all other higher bands, including the ~34 kDa band of the positive control, are contaminations. Protein quantification using Bradford indicated that the crude fraction contained 1.2 mg/mL total protein, while the 30–60% fraction contained 1.4 mg/mL protein, and the 40–50% fraction contained 0.9 mg/mL protein.

As the EIX protein is a xylanase enzyme and is used for its enzymatic activity in food and feed industries (Bhardwaj et al., 2019), we proceeded to assess whether the purified EIX is enzymatically functional. Xylanase activity in 30–60% and 40–50% ammonium sulfate fractions and the positive control was measured using the DNSA method with some modifications (see material and methods). The 30–60% and 40–50% fractions and the positive control had 4.07, 3.94, and 4.43 U activity, respectively (Figure 2; Table 1).

The enzyme precipitate obtained after 30–60% ammonium sulfate saturation was found to be purified 1.82-fold and possess 2.826 U/mg specific activity (Table 1), whereas the enzyme precipitate obtained with 40–50% ammonium sulfate saturation was purified 2.83-fold and had 4.377 U/mg specific activity. The crude extract had a specific xylanase activity of 1.546 U/mg. The positive control (EIX) had a specific activity of 3.878 U/mg. Ammonium sulfate precipitation purified the xylanase. The precipitated fractions had specific activity close to that of the positive control EIX, which was purified as previously described by Dean and Anderson (1991).

EIX is used as an elicitor of plant defense responses in plant immunity studies [10,11,14,30,31,32]. The main impetus for this work was the lack of easily obtainable immunogenic xylanase required for plant immunity research. Previous attempts to purify immuno-active xylanase from a bacterial system were unsuccessful, asthe fungal xylanase was not immuno-active when expressed in bacterial systems (Adi Avni personal communication), possibly due to the mature protein requiring eukaryotic modifications. We obtained industrial xylanase from two different manufacturers: xylanase from *Trichoderma viride* (Cas No. 9025-57-4, 100,000 units/g) was obtained from Newdaystart, TL, China, and xylanase livestock food additive was obtained from Vega group, Hangzhou, China. These preparations were likely formulated with an unknown crystallizing agent, as they could not be re-suspended in plant-compatible buffers. Re-suspension proved impossible in anything except 500 mM NaOH, which resulted in the necrosis of plant tissues upon treatment. We also obtained recombinant fungal xylanase from sigma (Cat. No. X2753, Cas No. 37278-89-0, 2500 units/g). This product is based on Novozymes corp. Pentopan^TM^. Unfortunately, this xylanase proved to not be plant immunogenic (Figure 3). The ultimate defense response in plants is the hypersensitive response (HR), which is a form of programmed cell death that prevents the spread of disease to uninfected tissue [33]. Xyn X2753 did not elicit HR or ethylene production upon treatment of plant tissues (Figure 3).

We examined the ability of our newly purified EIX to elicit defense responses. We infiltrated tobacco leaves with various dilutions of the 30–60% fraction, as western blot analysis indicated it to be the fraction containing the maximum amount of EIX, and compared it to the positive control diluted 1:1000 (Figure 4a,c). Dilutions 1:10, 1:100, and 1:1000 showed statistically similar levels of HR as compared to the PC, while a dilution of 1:5000 was significantly lower (Figure 4c). We, therefore, continued with the dilution of 1:1000, which we found to act similarly to the positive control (diluted 1:1000) in eliciting HR. We infiltrated tobacco leaves with a dilution of 1:1000 of our ammonium sulfate saturation fractions and monitored for the development of HR. We compared the 0–30% and 30–60% fractions, and the 30–40%, 40–50%, and 50–60% sub-fractions. Both the 30–60% fraction and the 40–50% sub-fraction were able to mount HR to a similar level as compared to the positive control, while the other sub-fractions led to a weaker HR (Figure 4b,c). This result points to EIX being precipitated mainly in the sub-fraction of 40–50%, with minimal xylanase loss to the other sub-fraction.

Among the earliest plant defense responses is the ROS burst. When comparing the ROS burst upon elicitation with different xylanase fractions, we see that both the 30–60% fraction and the 40–50% sub-fraction are able to mount ROS burst to a slightly lower extent when compared with the PC (Figure 4d,e).

As EIX is an ethylene-inducing xylanase, we proceeded to check the ability of the different purification fractions to induce ethylene biosynthesis. Tobacco tissue that was treated with the crude culture filtrate yielded similar ethylene levels as compared to the negative control treated with water (Figure 3f). The 30–60% ammonium sulfate saturated fraction led to the induction of ethylene at similar levels as the PC (Figure 4f). The 40–50% fraction led to ethylene induction to a lower level as compared to the positive control (Figure 4f).

## 3. Discussion

Here, we provide a simple, cost-effective protocol to produce plant immunogenic fungal xylanase. The xylanase produced in this manner showed similar plant immunogenicity and xylanase activity to the control EIX when adjusted for protein content. Due to the current lack of cost-effective, commercially available sources of xylanases that retain plant immunogenic activity, we set out to produce an immunogenic xylanase from the source fungus, *Trichoderma*, designing a method that is easily adaptable for lab-based production if desired.

Possible disadvantages of SmF protein production are its high cost and complex downstream processing. In this study, we used a cost-effective and easy process for the production of plant immunogenic xylanase. It has been suggested that a substrate containing xylan is necessary for xylanase production because its product is used both as an inducer and a carbon source for the microbes [25]. However, the high cost of xylan has limited its use in xylanase production, and xylan is not currently readily available for purchase worldwide. The use of agro-waste as a substrate for fungal production of xylanase could reduce the cost of production and increase productivity. We used wheat bran as a carbon source which is used in the domestic food market, as well as being an industrial agro-waste [26]. Wheat bran is a natural inducer of fungal xylanase production, likely due to its cell wall polysaccharide composition, which has 40% xylan. The Hydrolysis product of wheat bran contains large amounts of soluble sugars such as glucose, xylose, arabinose, and galactose, which the microbes can utilize as a carbon source for growth [34].

We were able to show that xylanase produced using our method could effectively elicit plant immunity in a manner similar to the positive control xylanase.

The xylanase activity is not required for the elicitation of plant defense responses [29]. We therefore showed that our purified protein not only retains its immune eliciting activity, but its enzymatic activity as well. Both the 30–60% fraction and 40–50% sub-fraction have similar enzymatic activity as the positive control. Thus, we conclude that we were able to purify xylanase from *Trichoderma* at high amounts, ensuring that it retains both its immune eliciting activity and its enzymatic activity.

## 4. Materials and Methods

### 4.1. Microorganism and Culture Condition

The fungal strain *T. harzianum* T39, a naturally occurring strain originally isolated from cucumber fruit [35], was obtained from Yigal Elad [36]. The culture was maintained on Potato dextrose agar (PDA, Difco BD) at 28 °C.

### 4.2. Production of Xylanase from T. harzianum

Xylanase from T39 was produced under submerged culture conditions using media comprised of wheat bran 20.0 g/L and salt solution (4.0 g/L each of K_2_HPO_4_, KH_2_PO_4_, and NH_4_SO_4_). These conditions were previously examined in connection with the purification of exo-polygalacturonase from *Aspergillus niger* [37]. We selected the purification conditions most suitable for downstream processing. Six 500 mL Erlenmeyer flasks, each containing 100 mL submerged medium, were inoculated with a single *Trichoderma* mycelium plug. The flasks were kept at 28 °C and 200× rpm in an incubator shaker. Maximum production of the enzyme was obtained 96 h after inoculation.

### 4.3. Secreted Protein Extraction

On the fifth day, the media was filtered through four layers of cheesecloth, and the filtrate was centrifuged at 10,000× rpm for 20 min. The supernatant that was obtained was filtered using Whatman filter paper 1. The filtrate obtained was used for enzyme purification.

### 4.4. Enzyme Purification

The crude samples were fractionated in different ranges of ammonium sulfate saturation: 0–30%, 30–60%, and 60–90%. During ammonium sulfate precipitation, the salt was added in small quantities under constant stirring at 25 °C. Salt was first added at up to 30% saturation. The treated crude enzyme solution was allowed to stand overnight at 4 °C and centrifuged at 10,000× rpm for 15 min. The pellet was discarded, and the supernatant was saturated up to 60% with ammonium sulfate and allowed to stand overnight. The supernatant obtained after centrifugation at 10,000× rpm was again saturated with ammonium sulfate up to 90% and was kept undisturbed at 4 °C overnight. This was centrifuged at 10,000× rpm for 15 min and the pellet obtained was dissolved in 3 mL of cold distilled water. Similarly, the crude samples were also fractioned to ammonium sulfate saturation 0–30%, 30–40%, 40–50%, and 50–60%. The dissolved pellets were dialyzed to remove excess salt and small proteins. Dialyzed enzyme fractions were used for protein estimation ([38] and SDS-PAGE), xylanase activity assays, and plant defense assays.

### 4.5. Coomassie Staining and Western Blot

Samples were boiled after adding sample buffer (8% SDS, 40% glycerol, 200 mM Tris-Cl, pH 6.8, 388 mM dithiothreitol (DTT), and 0.1 mg/mL bromophenol blue dye). Samples were run in 16% sodium dodecyl sulfate-polyacrylamide gel electrophoresis. For coomassie staining, the gel was submerged in staining solution (0.125% coomassie blue R-250, 10% isopropanol, 10% acetic acid) and later de-stained using 5% methanol and 7.5% acetic acid. For Western blot analysis, the gel was blotted onto nitrocellulose membranes and incubated with EIX-specific polyclonal antibodies [29].

### 4.6. Xylanase Assay

Enzyme activity of xylanase was assayed by determining the liberated reducing end products by standard method (Miller 1959) with some modifications. The reaction solution (2 mL) consisted of 0.5 mL of 1% birchwood xylan, 1.4 mL 100 mM phosphate buffer (pH 6.5) and 0.1 mL enzyme solution. It was incubated for 20 min at 37 °C in a water bath. Three mL of dinitrosalicyclic acid (DNSA) reagent was added, and the volume was brought to 6 mL by the addition of 1 mL of distilled water. The solution was boiled for 10 min in a water bath, cooled, and absorbance was read at 540 nm. A negative control was simultaneously prepared using a thermally denatured enzyme. The concentration of the product (xylose) was determined with the help of a calibration curve. One unit of xylanase activity is defined as the amount of enzyme that liberates 1 μmol of xylose per min per unit volume under the assay conditions.

### 4.7. ROS Measurement

ROS burst was measured as previously described by Leibman-Markus et al. [39]. Leaf disks (0.5 cm in diameter) were harvested from tomato plants (cv M82). Disks were floated in 250 μL of ddH_2_O in a white 96-well plate (SPL Life Sciences, Korea) overnight at room temperature. After incubation, the water was removed, ROS measurement solutions containing the relevant xylanase fractions were added, and light emission was immediately measured for a period of 45 min, using a microplate luminometer (GloMax^®^ Discover, Promega, Madison, WI, USA).

### 4.8. Ethylene Measurement

Ethylene biosynthesis was measured as previously described by Leibman-Markus et al., 2017. Leaf disks (0.9 cm in diameter) were harvested from tobacco plants (*Nicotiana tabaccum* cv Samsun NN). Six leaf disks were sealed in 15 mL glass tubes containing 1 mL of assay medium (250 mM sorbitol, 10 mM MES, pH 5.8) and 1 μL of the different xylanase fractions, and incubated with shaking overnight at room temperature. Ethylene production was measured by gas chromatography (Varian 3300, Varian, CA, USA).

## 5. Conclusions

The method described here offers a rapid, cost-effective, and reproducible process for producing xylanase from *Trichoderma*, easily executed in research lab settings. The production and purification method is a simple one and supports the abundant production of plant immunogenic xylanase that retains enzymatic activity. In addition to providing a simple and cost-effective method for producing plant immunogenic microbial enzymes in research settings, the described method may also provide an alternative for other fungi, in which the induction of secretory enzymes is difficult.

## Figures and Tables

**Figure 1 ijms-22-04214-f001:**
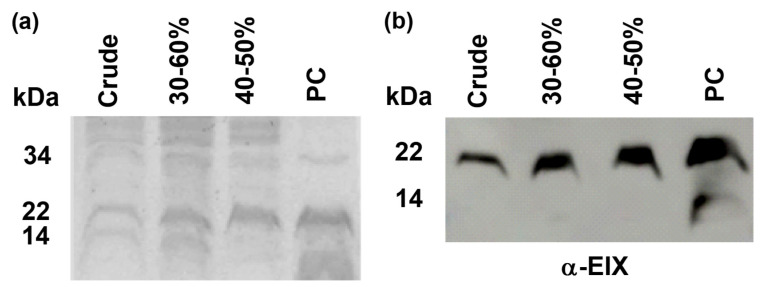
Verification of Ethylene Inducing Xylanase (EIX) presence and evaluation of purification level. Equal amounts of extracts and positive control were loaded onto 16% SDS-PAGE. (**a**) Coomassie staining. Gel was submerged in staining solution for 30 min and later de-stained overnight. (**b**) Western blot. Gel was blotted onto a nitro-cellulose membrane and subsequently incubated with polyclonal anti-EIX antibodies Adapted from [29].

**Figure 2 ijms-22-04214-f002:**
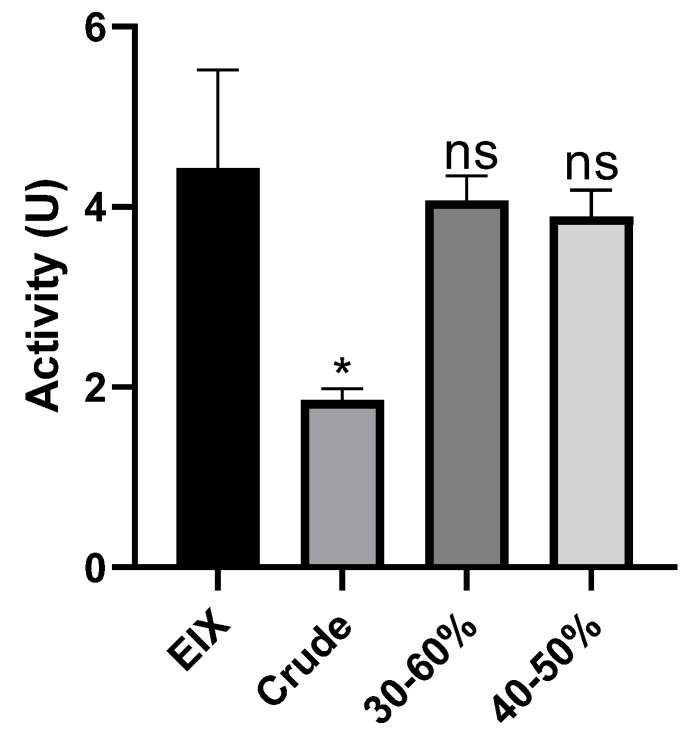
Xylanase activity. Xylanase activity in Ethylene Inducing Xylanase (EIX; positive control), crude, 30–60% and 40–50% ammonium sulfate fractions, was measured by the DNSA method (see Materials and methods). Error bars represent the average ± SEM of three independent experiments, asterisks represent significant differences: * *p* < 0.05, ns = non-significant (two-tailed t-test).

**Figure 3 ijms-22-04214-f003:**
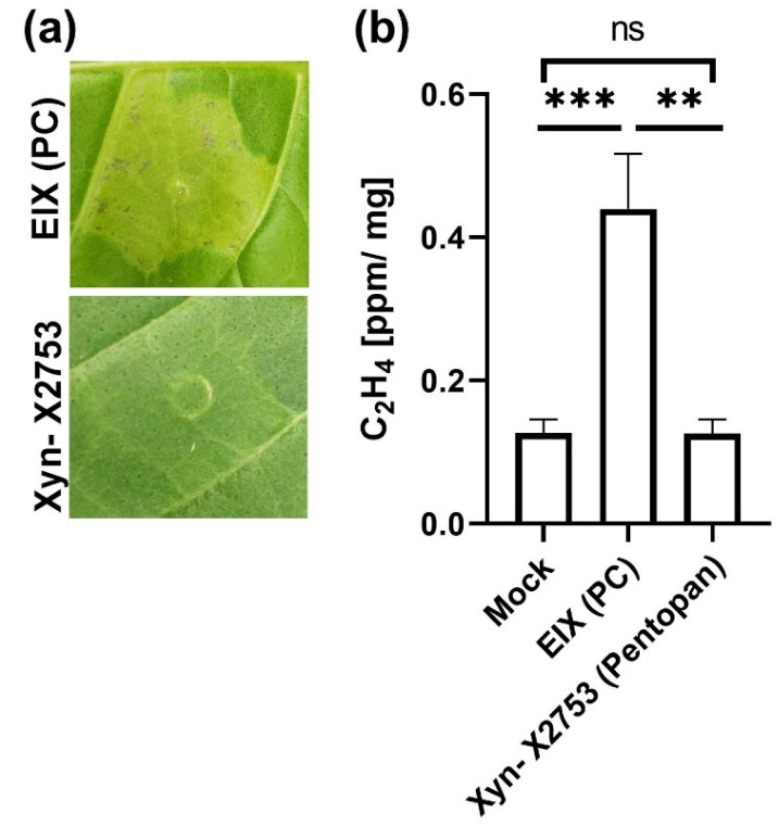
A commercially available xylanase does not elicit plant defense. (**a**) Hypersensitive Response. Leaves of tobacco plants were injected with 1 ug/mL Ethylene Inducing Xylanase (EIX) or the Xylanase Xyn- X2753. HR development was photographed 72 h post injection. (**b**) Ethylene biosynthesis. Leaf disks of 5 week old tobacco plants were floated overnight on a 250 mM of sorbitol solution with 1 μg/mL EIX or Xyn- X2753. Ethylene induction is presented as ppm ethylene/mg tissue. Error bars represent the average ± SEM of three independent experiments, asterisks represent significant differences: ** *p* < 0.01, *** *p* < 0.001, ns = non-significant (*n* = 5, one-way ANOVA, *p* < 0.0015).

**Figure 4 ijms-22-04214-f004:**
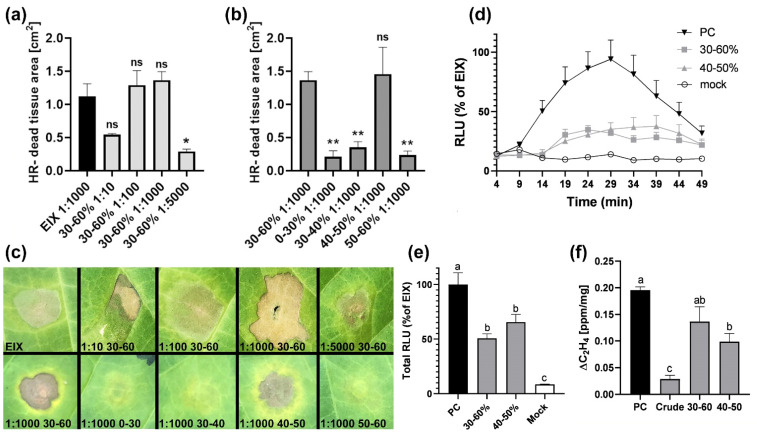
Elicitation of plant defense responses by purified Ethylene Inducing Xylanase (EIX) fractions. (**a**–**c**) Hypersensitive Response. Leaves of tobacco plants were injected with (**a**). Serial dilutions of fraction 30–60%. (**b**) Equal amounts of different fractions as indicated. (**c**) HR development was photographed 72 h post injection. Quantification of results from two biological repeats ±SE, *n* ≥ 3. Asterisks (differences between positive control and treatment) indicate significance in one-way ANOVA with a Bonferroni post hoc test, * *p* < 0.05, ** *p* < 0.01, ns = non-significant. (**d**,**e**) Reactive oxygen species measurements. Leaf disks of M82 plants were floated on water overnight and then replaced with luminescence solution containing indicated purification fractions. Luminescence (RLU) was immediately measured to track the ROS burst. Average ± SEM values of three independent experiments, *n* = 12 each. (**d**) kinetic graph. (**e**) Summary of total RLU. Letters represent significant differences (one-way ANOVA, *p* < 0.01). (**f**) Ethylene biosynthesis. Leaf disks of 5-week old tobacco plants were floated overnight on a 250 mM of sorbitol solution with indicated fractions. Ethylene induction was defined as the ΔEthylene (Treatment_Ethylene_–Mock_Ethylene_). Error bars represent the average ± SEM of three independent experiments, letters represent significant differences (*n* = 21, one-way ANOVA, *p* < 0.05).

**Table 1 ijms-22-04214-t001:** Purification table of Ethylene Inducing Xylanase (EIX) produced from *T. harzianum* using submerged fermentation.

Fraction	Total Activity (U)	Total Protein (mg)	Specific Activity (U/mg)	Purification Fold
Crude	371.2 ± 25.3	240	1.546	-
30–60%	40.7 ± 2.7	14.4	2.826	1.82
40–50%	39.4 ± 2.9	9.0	4.377	2.83

## Data Availability

The authors declare that the data supporting the findings of this study are available within the paper and its supplementary information files. Raw data is available from the corresponding author upon reasonable request.

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
