# Peer review of "Method for the Production and Purification of Plant Immuno-Active Xylanase from Trichoderma"

_ijms, 2021, doi:10.3390/ijms22084214_

Round 1
Reviewer 1 Report
Dear Authors,
I have an honor to review manuscript entiled: „ Method for the production and purification of plant immuno- active Xylanase from Trichoderma” submitted to IJMS MDPI.
Authors presented interesting and promising findings concentrated on producing xylanases by filamentous fungi and posibilities of secreting them in liquid cultures, making some kind of system for xylanase purification. Authors postulated that they describe a method for cost and yield effective production of xylanase from Trichoderma using wheat bran as a growth substrate. Moreover, xylanase activity and effectively induced plant immunity, what was tested by ethylene biosynthesis, ROS burst.
Despite interesting new findings, I have some specific comments to author’s statements.
- I can’t agree with statement, that authors ‘tested plant immunity’ thus hypersensitive response’ – hypersensitive response is a type of immunity, as well as programmed cell death. So, it is unable to write „method possessed xylanase activity and effectively induced plant immunity as tested by ethylene biosynthesis, ROS burst and hypersensitive response” -in abstract and introduction;
- In material and methods sections- „EIX-specific polyclonal antibodies” please, add specification of used antibodies [not only producer, but also what kind of of sequence these antibodies recognised;
- I have a question, to ROS burst detection, what kind of reactive oxygen species was recognised in these methods;
- Figure 4 is almost unreadible - please, enlarged it, especially: a,b,d,e,f – of course, please, safe an adequete resolution;
- I suggest to underlined morethe role of production of plant immunogenic xylanase in conclusions;
Author Response
Many thanks to the Reviewers for their time and efforts, which have improved the manuscript. Follows a "point-by-point" response to the reviewer comments.
Dear Authors,
I have an honor to review manuscript entiled: „ Method for the production and purification of plant immuno- active Xylanase from Trichoderma” submitted to IJMS MDPI.
Authors presented interesting and promising findings concentrated on producing xylanases by filamentous fungi and posibilities of secreting them in liquid cultures, making some kind of system for xylanase purification. Authors postulated that they describe a method for cost and yield effective production of xylanase from Trichoderma using wheat bran as a growth substrate. Moreover, xylanase activity and effectively induced plant immunity, what was tested by ethylene biosynthesis, ROS burst.
Despite interesting new findings, I have some specific comments to author’s statements.
- I can’t agree with statement, that authors ‘tested plant immunity’ thus hypersensitive response’ – hypersensitive response is a type of immunity, as well as programmed cell death. So, it is unable to write „method possessed xylanase activity and effectively induced plant immunity as tested by ethylene biosynthesis, ROS burst and hypersensitive response” -in abstract and introduction;
The hypersensitive response is considered a facet of the plant immune response, which is not always activated, e.g., it is not activated in response to flg-22 or in incompatible interactions. We have amended the abstract and introduction as suggested by the reviewer (see lines 24-26 and 68-70).
- In material and methods sections- „EIX-specific polyclonal antibodies” please, add specification of used antibodies [not only producer, but also what kind of of sequence these antibodies recognised;
We included a reference for the antibodies, which were produced in the Avni lab at Tel-Aviv University, and are not commercial antibodies. See also the acknowledgment section. By nature, polyclonal antibodies do not recognize one specific sequence but rather, a variety of epitopes. The sequences recognized by the antibody we used have not been elucidated, though a consensus sequence suggested to be recognized by some of these antibodies is provided in the paper we cited.
- I have a question, to ROS burst detection, what kind of reactive oxygen species was recognised in these methods;
The Horse Radish Peroxidase/ luminol method we used to measure ROS is thought to mostly recognize hydrogen peroxide (H2O2) produced by NADPH oxidase in response to pathogen attack or elicitation, as detailed in the reference we cited.
- Figure 4 is almost unreadible - please, enlarged it, especially: a,b,d,e,f – of course, please, safe an adequete resolution;
Amended with thanks.
- I suggest to underlined more the role of production of plant immunogenic xylanase in conclusions;
Amended with thanks (lines 297-301 and 326).
Reviewer 2 Report
As reported by authors, manuscript describe a method for cost and yield effective production of xylanase by Trichoderma T39. I found no particular issues in present paper because procedures are well described and results are clear.
As a possible improvement for manuscript, I would appreciate to know if other growth conditions or substrates were tested and the related yields. These information could be of interest for other researchers.
Best regards
Author Response
Many thanks to the Reviewers for their time and efforts, which have improved the manuscript. Follows a "point-by-point" response to the reviewer comments.
"As reported by authors, manuscript describe a method for cost and yield effective production of xylanase by Trichoderma T39. I found no particular issues in present paper because procedures are well described and results are clear.
As a possible improvement for manuscript, I would appreciate to know if other growth conditions or substrates were tested and the related yields. These information could be of interest for other researchers."
Many thanks for these comments. We optimized the method for growing the fungus, i.e., conditions and substrates, by selecting the purification method most suitable for downstream processing, according to Anand et al., 2017. Maximal extracellular protein content was obtained after 96 h, as detailed in the manuscript. We added a comment to this effect in the text (lines 112-114).
Anand G, Yadav S, Yadav D. 2017. Production, purification and biochemical characterization of an exo-polygalacturonase from Aspergillus niger MTCC 478 suitable for clarification of orange juice. 3 Biotech 7: 1–8.
Reviewer 3 Report
Manuscript ijms-1155987 refers to the production and purification of a xylanase elicitor. Manuscript is suitable for publication. I have only minor comments / suggestions.
Key-words. I suggest using ”innate immunity” and not ”plant immunity”. MAMPs are elicitors of innate immunity in the
L40. I would suggest to not use inducers as a synonym for elicitor. To elicit means "to draw out something hidden, latent, or reserved,"; to induce mean ”to cause something to happen”. It is a subtle difference, elicitor action is directed related to the capacity of the host to determine an immune response.
L105. I suggest to mention that strain T39 is a naturally occurring strain.
Author Response
Many thanks to the Reviewers for their time and efforts, which have improved the manuscript. Follows a "point-by-point" response to the reviewer comments.
Comments and Suggestions for Authors
Manuscript ijms-1155987 refers to the production and purification of a xylanase elicitor. Manuscript is suitable for publication. I have only minor comments / suggestions.
Key-words. I suggest using ”innate immunity” and not ”plant immunity”. MAMPs are elicitors of innate immunity in the
Amended with thanks (line 27).
L40. I would suggest to not use inducers as a synonym for elicitor. To elicit means "to draw out something hidden, latent, or reserved,"; to induce mean ”to cause something to happen”. It is a subtle difference, elicitor action is directed related to the capacity of the host to determine an immune response.
Amended with thanks (lines 41-42).
L105. I suggest to mention that strain T39 is a naturally occurring strain.
T39 was originally isolated from cucumber. We added the information to the manuscript (lines 106-107).